# A gate-free monolayer WSe$_2$ pn diode

Jhih-Wei Chen[1], Shun-Tsung Lo[1], Sheng-Chin Ho[1], Sheng-Shong Wong[1], Thi-Hai-Yen Vu[1], Xin-Quan Zhang[2], Yi-De Liu[1], Yu-You Chiou[1], Yu-Xun Chen[3], Jan-Chi Yang [1], Yi-Chun Chen[1], Ying-Hao Chu[4], Yi-Hsien Lee [2], Chung-Jen Chung[5], Tse-Ming Chen[1,5], Chia-Hao Chen[6] & Chung-Lin Wu[1,6]

Interest in bringing p- and n-type monolayer semiconducting transition metal dichalcogenides (TMD) into contact to form rectifying pn diode has thrived since it is crucial to control the electrical properties in two-dimensional (2D) electronic and optoelectronic devices. Usually this involves vertically stacking different TMDs with pn heterojunction or, laterally manipulating carrier density by gate biasing. Here, by utilizing a locally reversed ferroelectric polarization, we laterally manipulate the carrier density and created a WSe$_2$ pn homojunction on the supporting ferroelectric BiFeO$_3$ substrate. This non-volatile WSe$_2$ pn homojunction is demonstrated with optical and scanning probe methods and scanning photoelectron microspectroscopy. A homo-interface is a direct manifestation of our WSe$_2$ pn diode, which can be quantitatively understood as a clear rectifying behavior. The non-volatile confinement of carriers and associated gate-free pn homojunction can be an addition to the 2D electron–photon toolbox and pave the way to develop laterally 2D electronics and photonics.

[1] Department of Physics, National Cheng Kung University, Tainan 70101, Taiwan. [2] Department of Materials Science and Engineering, National Tsing Hua University, Hsinchu 30013, Taiwan. [3] Department of Electrophysics, National Chiao Tung University, Hsinchu 30010, Taiwan. [4] Department of Materials Science and Engineering, National Chiao Tung University, Hsinchu 30010, Taiwan. [5] Center for Micro/Nano Science and Technology, National Cheng Kung University, Tainan 70101, Taiwan. [6] National Synchrotron Radiation Research Center (NSRRC), Hsinchu 30076, Taiwan. Correspondence and requests for materials should be addressed to C.-H.C. (email: chchen@nsrrc.org.tw) or to C.-L.W. (email: clwuphys@mail.ncku.edu.tw)

D evices that require low power consumption, materials that have a monolayer structure with quantum confinement, electrons and holes that convey information with high mobility—these are just some breakthroughs that might be realized following the development of two-dimensional (2D) materials that are efficient, scalable, and easily engineered to achieve diverse functionality. Since the discovery of various 2D materials almost a decade ago, the recent boost of interest in semiconducting layered transition metal dichalcogenides (TMD) originates from their exotic characteristics in the monolayer limit, such as giant spin-valley coupling[1,2], optical control of valley polarization and coherence[3,4], an indirect-to-direct bandgap transition[5–7] and tightly bound excitonic states[8–10]. Despite these fantastic discoveries, the present challenge is to promise a TMD pn diode, which is a fundamental building block of modern devices, that manifests all their numerous advantages (including monolayer structure, homo-interface, and device functionality) while also easily overcoming the scaling limit of current complementary metal-oxide semiconductor (CMOS) technology or achieving atomically thin optoelectronics. The methods used to achieve p- and n-doping in TMD are mainly inducing charge transfer to TMD, such as gate-bias tuning[11–13], interacting with atoms/molecules[14–17], molecular adsorption on a surface[18,19], and plasmonic hot-electron doping[20,21]. The method most commonly used to dope TMD is to set a gate voltage through the metal gating, which provides a direct way to realize the strong charge-density tuning, but metal gates would result in an inhomogeneous charge distribution and unavoidable degradation of the emission of light at the TMD surface. TMD diodes have been constructed by stacking vertically, such as an ionic liquid-gated bulk $MoS_2$ device[22], a TMD/III–V semiconductor[23] and a TMD/doped-silicon[24]. TMD heterojunctions vertically involving other materials, however, apparently lack many appealing exotic properties of a lateral monolayer.

Because of the ultra-thin nature of a TMD, a necessary substrate provided by a functional material can offer a strategy for lateral modulation of the TMD band structure without problems caused by the doping defects and the mismatches between the dopants and 2D lattice atoms. Ferroelectric (FE) materials possess a spontaneous electrical polarization that can be macroscopically and locally inverted with an external stimulus, and are considered to be prospective substrates to support TMD to achieve a pn homojunction, as the accumulation or depletion of an inevitably charged mobile carrier occurs in the TMD to screen the polarization field of the FE substrate. Here, using detailed spatially resolved spectroscopies, we have demonstrated the respective $WSe_2$ electron-filling (n-type) and electron-emptying (p-type) regions configured and modified with the FE domains of a $BiFeO_3$ (BFO) substrate, and thus define a monolayer $WSe_2$ pn homojunction (sketched in Fig. 1). The current flowing through a $WSe_2$ pn diode is non-volatilely rectified with a source-drain voltage ($V_{SD}$) without assistance of gate biasing. This diode shows a strong current-rectifying behavior in electrical transport properties, which confirms the results revealed in the homojunction band structure. This work provides a non-volatile control of TMD doping and a promising way to produce a pn homojunction as a future building block of 2D device applications.

## Results

**Scanning probe microscopy and μ-PL characterization**. The crystalline ferroelectric BFO layers were grown with pulsed-laser deposition (PLD) on (001) $SrTiO_3$ (STO) substrates with a conductive $SrRuO_3$ (SRO) layer (Methods section). Using chemical-vapor deposition (CVD) and a wet transfer method, we affixed monolayer crystalline $WSe_2$ sheets firmly to the BFO surface,

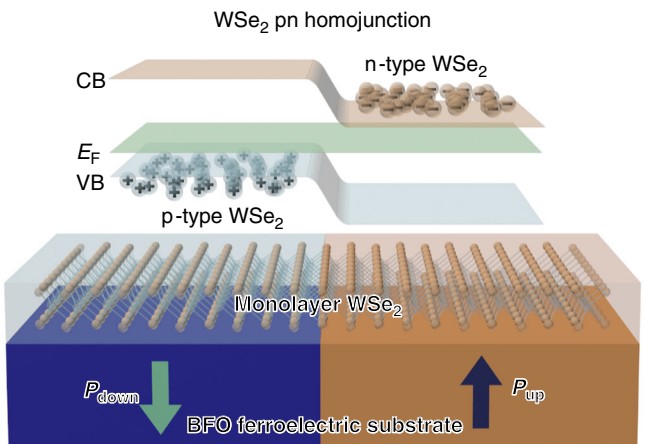

**Fig. 1** Schematic band diagram of a $WSe_2$ pn homojunction derived from a ferroelectric-pattern-assisted BFO layer. Both polarization states ($P_{down}$ and $P_{up}$) on a ferroelectric BFO layer can directly affect the carrier type of monolayer TMD with either p-type or n-type semiconducting behavior

forming a van der Waals (vdW) interface that played a key role in the formation of a $WSe_2$ homojunction. The scanning-probe characterizations and images of a representative $WSe_2$ on ferroelectrically patterned BFO are displayed in Fig. 2. As in the case of the scanning line profile, the thickness of the $WSe_2$ was a monolayer (~1.5 nm) measured by tapping-mode atomic force microscope (AFM) and confirmed by photoluminescence (PL). The monolayer thickness of $WSe_2$ measured here was larger than the mechanically exfoliated monolayer $WSe_2$ ($d_{WSe2}$ ~ 0.7 nm)[19] and the CVD-grown monolayer $WSe_2$ ($d_{WSe2}$ ~ 1.1 nm)[25,26] since the water molecules were easily trapped on the hydrophilic $BiFeO_3$ surface to increase the distance between $WSe_2$ and $BiFeO_3$ substrate[27,28] and thus increase the thickness measured by AFM. The ferroelectric properties of the BFO layer were verified through characterization with polarization versus voltage ($P–V$) and a piezo-force microscope (PFM) in Supplementary Figure 1 and Fig. 2b, respectively. The $P–V$ loops show that the BFO films used in this work having naturally downward polarization ($P_{down}$) undergo sharp FE switching during poling, with a remnant polarization ≈60 μC cm$^{-2}$ (Supplementary Figure 1)[29]. To demonstrate the ferroelectric control of $WSe_2$-doped charges, we created a ferroelectric domain pattern on reversing the polarization through scanning with a metal probe (probe voltage set to −8 V) to obtain an area of upward polarization ($P_{up}$) that is partially covered with $WSe_2$; the overlap area is about 7 μm². In the PFM image shown in Fig. 2b, two distinct $P_{up}$ and $P_{down}$ regions are revealed under a $WSe_2$ sheet with opposite out-of-plane phases; the shape of the $WSe_2$ sheet is consistent with that measured from an AFM image, implying that the $WSe_2$ sheet is not structurally damaged during the reversal of polarization.

To reveal the substrate-induced charge traps in $WSe_2$, which can significantly alter the surface potential and work function of $WSe_2$, measurements with a Kelvin probe microscope (KPM) showed that the surface potential of $WSe_2$ decreases about 450 meV because of the induced charge screening in $WSe_2$ within the $P_{up}$ area, compared within a $P_{down}$ area as shown in Fig. 2c. This potential difference of $WSe_2$ (~0.45 eV) created with ferroelectric $P_{up}$ and $P_{down}$ regions of BFO is significantly larger than with an epitaxial growth of monolayer $WSe_2$-$MoS_2$ lateral junction[30], which shows that the ferroelectricity of the supporting substrate has the ability to affect the electrical properties of TMD material efficiently and strongly. Moreover, the monolayer $WSe_2$ has a direct band gap that offers a large quantum yield of radiative exciton recombination, leading to efficient PL emission. These

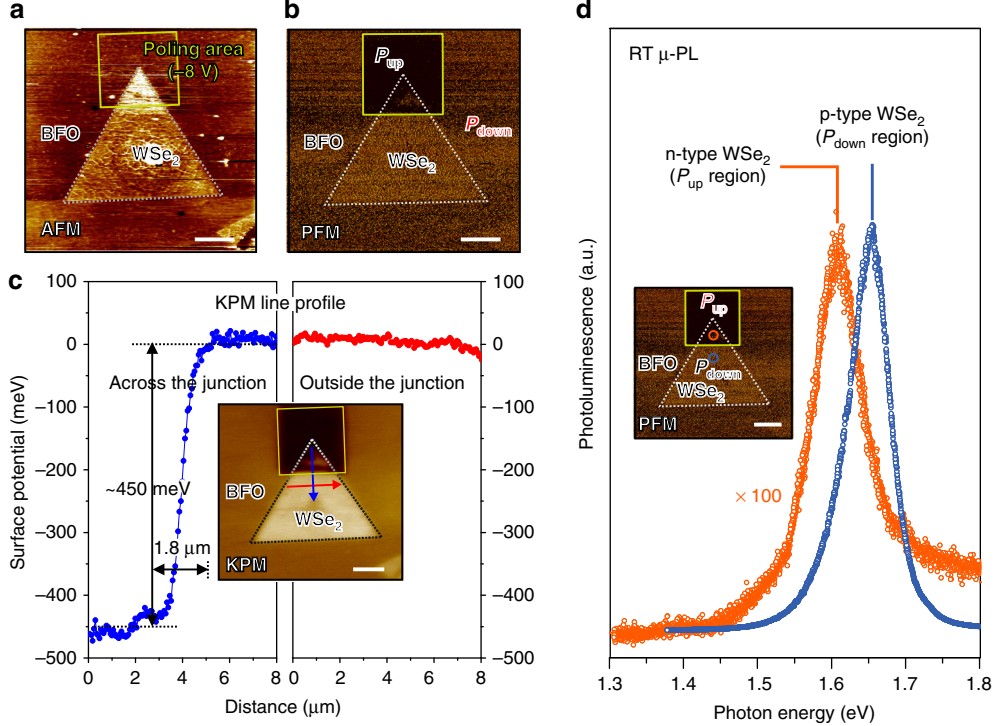

**Fig. 2** The scanning probe microscope images and μ-PL spectra of the WSe$_2$ pn homojunction. WSe$_2$ on $P_{up}$ and $P_{down}$ regions from measurements with an AFM and PL. **a, b** AFM and PFM taken of a WSe$_2$ sheet on the BFO ferroelectric layer as grown. The solid line shows poled regions with bias −8 V at an AFM tip (yellow lines). A PFM image taken of a WSe$_2$ sheet on a BFO layer with $P_{up}$ and $P_{down}$ regions, which shows the out-of-plane ferroelectric polarization in BFO to have phase difference 180°. **c** KPM image and line profile image taken across and outside the $P_{up}$ and $P_{down}$ homojunction. **d** PL of a WSe$_2$ sheet taken from the $P_{up}$ and $P_{down}$ homojunction. The scale bars in the figures are 5 μm

photo-excited electrons and holes can form either excitons (e–h pairs) or electron-/hole-bounded trions (e–e–h/e–h–h) even near 300 K, and, accordingly, can be used to monitor the carrier type (doping type) and density in a TMD having direct bandgap emission[31]. The spatially resolved PL spectra recorded from the same area (circle in Fig. 2d) of the monolayer WSe$_2$ sheet at two opposite polarization states is depicted in Fig. 2d. It is clearly visible that the pronounced PL emission induced a red shift in position and a significant attenuation of intensity from the natural $P_{down}$ state (~1.65 eV) to the reversed $P_{up}$ state (~1.60 eV). This is consistent with the fact that the emission switched to decreased energy and quantum yield from a hole-bounded trion recombination to an electron-bounded trion recombination in the initially p-type WSe$_2$ and then becoming electron-accumulated WSe$_2$ under the $P_{up}$ state of the BFO substrate, which is in agreement with previous reports on the PL characteristics of monolayer TMD under electrostatic gating[32]. Moreover, we recorded PL spectra in a series at the same spot on the sheet under another FE switching cycle, which confirmed that the attenuation of the PL emission is not from destruction due to biased tip scanning during FE poling, as shown in Supplementary Figure 2.

**The SPEM/S measurements**. To directly visualize the ferroelectric tuning in the electronic structure of WSe$_2$ on a BFO, we used a scanning photoelectron microscope and spectroscopy (SPEM/S, in National Synchrotron Radiation Research Center (NSRRC), Hsinchu, Taiwan) to provide the required spatial resolution (~sub-μm) for chemical mapping and the energy resolution (± 50 meV) for localized photoelectron spectroscopy (μ-PES) with a beam of synchrotron radiation (SR) focused on the $P_{up}$ and $P_{down}$ regions larger than the size of the focused SR beam. Figure 3a, b shows Se 3$d$, W 4$f$, and Bi 4$f$ core-level

photoelectron spectra, which correspond to the SPEM images taken from a WSe$_2$ sheet at the $P_{up}$ and $P_{down}$ regions, respectively. The binding energy (BE) of the core-level electron in Se 3$d$ and W 4$f$ of the $P_{up}$ region was significantly greater than in $P_{down}$, but was not observed in Bi 4$f$. For the band structure of the $P_{down}$ region sketched in Fig. 3c, the entire Fermi-level energy is located 0.4 eV above the valence band, which displays a p-type semiconductor behavior. An as-grown $P_{down}$ BFO layer hence preserved the p-type behavior, because p-type doped BFO thin film has a work function value near that of WSe$_2$[33,34]. The detailed band structures deduction of WSe$_2$ at $P_{up}$ and $P_{down}$ are shown in Supplementary Figures 3 and 4. Explicitly, with ferroelectric polarization switching, we observed that Se 3$d$ and W 4$f$ have a BE shift ~1.0 eV in the $P_{up}$ region. This energy shift is observed also in a SPEM image, which reveals a contrast reversal in the W 4$f$ images. The energy difference of the core level corresponds to the Fermi-level energy shift in the band gap, which turned the p-type into n-type WSe$_2$, as shown in Fig. 3c. The tuning of the Fermi level ($E_F$) within the gap that is significantly larger than the previously reported number for Nb-doped MoSe$_2$ and NO$_x$-doped WSe$_2$[14,16] implies that it is possible to tune the $E_F$ position near the TMD band edge with heavily doping according to this approach. Here, this ferroelectricity-assisted band structure engineering results a WSe$_2$ pn homojunction at room temperature.

To determine the ferroelectricity-induced manipulation of WSe$_2$ carrier density, the PES spectra of WSe$_2$ under $P_{up}$ and $P_{down}$ BFO states are required to reveal the respective $E_F$ energies relative to their valence ($E_V$) and conduction ($E_C$) band edges. Under a parabolic approximation for the band dispersion near the bottom of conduction band (CB) and the top of valence band (VB) modeled in the effective mass of mobile carrier with Fermi–Dirac statistics, the 2D electron density ($\sigma_n$) in WSe$_2$ is

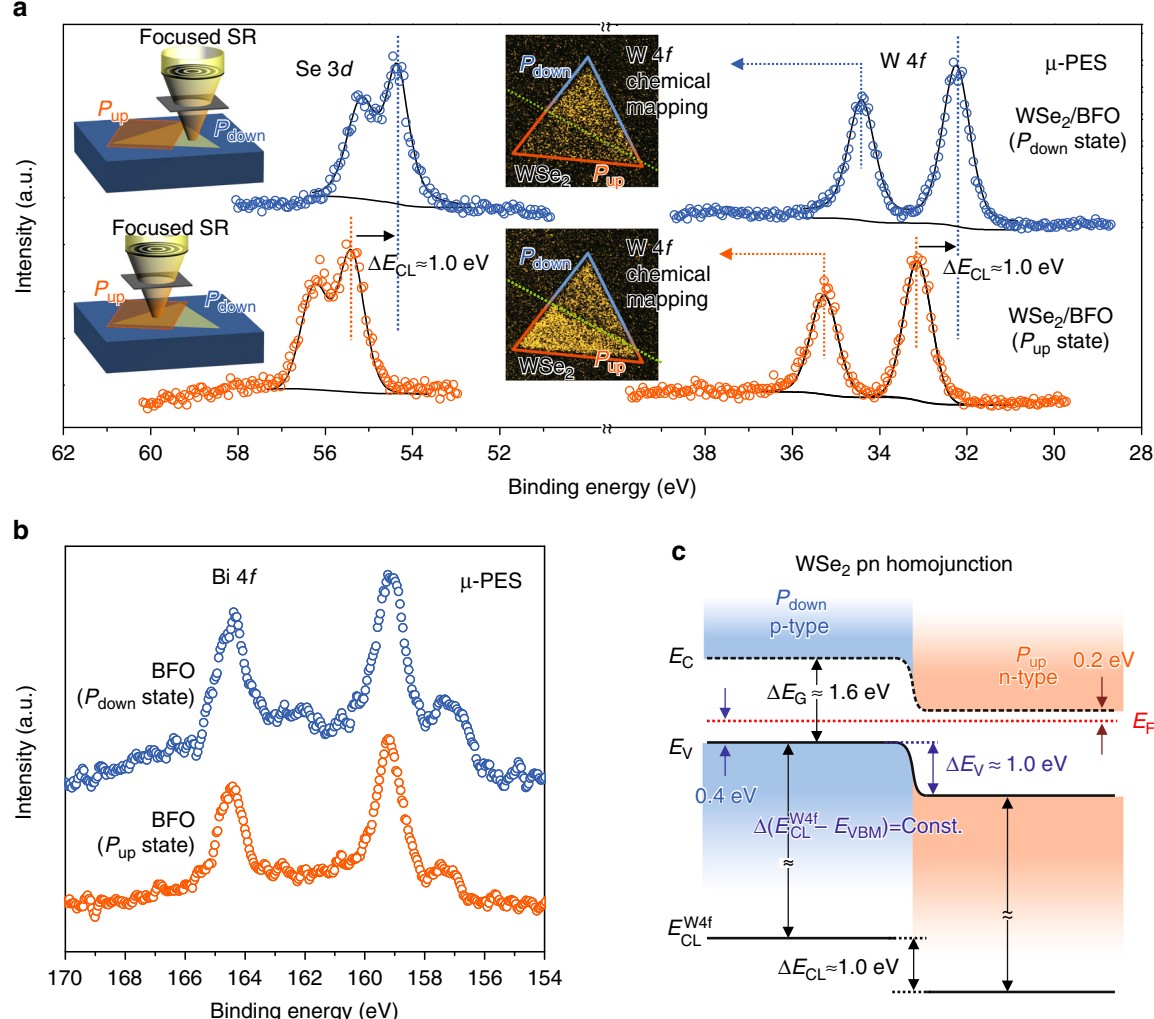

**Fig. 3** SPEM images and μ-PES measurements on the WSe$_2$ pn homojunction. Se $3d$, W $4f$, and Bi $4f$ core-level photoelectron spectra measured with SPEM in $P_{up}$ and $P_{down}$ regions of a WSe$_2$/BFO homojunction. **a** Core-level spectra of Se $3d$ and W $4f$ recorded from a $P_{down}$ (blue) and a $P_{up}$ (orange) region. SPEM images of W $4f$ taken in 34.6 eV and 35.4 eV, which correspond to a $P_{down}$ and a $P_{up}$ region, respectively. **b** Core-level spectra of Bi $4f$ emitted from the BFO substrate. **c** The band structure deduced from **a** reveals the pn junction for WSe$_2$ in $P_{down}$ and $P_{up}$ regions near 300 K

given by $\sigma_n = (g_{2D}k_BT)\ln\{1 + \exp[(E_F-E_C)/k_BT]\}$ and the 2D hole density ($\sigma_p$) in WSe$_2$ is $\sigma_p = (g_{2D}k_BT)\ln\{1 + \exp[-(E_F-E_V)/k_BT]\}$, in which $g_{2D}$ is the electron/hole density of state for WSe$_2$. As shown in Fig. 3c, the $E_F-E_V$ of WSe$_2$ is about 0.4 eV in the $P_{down}$ state and $E_C-E_F$ of WSe$_2$ is about 0.2 eV in the $P_{up}$ state, which are determined by the spectra shown in the Supplementary Note 3. As a result, the carrier densities in the p-doped and n-doped WSe$_2$ regions are estimated of $\sigma_p \sim 8.63 \times 10^9$ cm$^{-2}$ and $\sigma_n \sim 1.48 \times 10^{13}$ cm$^{-2}$, respectively, and the detailed calculation is provided in the Supplementary Note 6. Comparing the huge electron density tuning range ($\sim 10^{10}$ cm$^{-2}$) based on the intrinsic carrier density ($\sigma_i \sim 1.26 \times 10^3$ cm$^{-2}$) of WSe$_2$, the mobile hole accumulation in WSe$_2$ is slightly inhibited ($\sim 10^7$ cm$^{-2}$) when the polarization in BFO layer is naturally $P_{down}$, which is in agreement with the results of the defect charge screening of naturally polarization field ($P_{down}$) in BFO proposed in previous results[27]. Due to the large polarization field provided by a ferroelectric BFO substrate, the large improvement of the mobile electron density ($\sim 10^{13}$ cm$^{-2}$) in TMD system observed in this study agrees satisfactorily with the order of surface bound charges of the BFO layer ($\sim 10^{14}$ cm$^{-2}$) and is higher than other TMD junction systems that was reported for elemental doping MoSe$_2$ ($\sigma_n \sim 10^{11}$ cm$^{-2}$) and lateral heterojunction WSe$_2$/MoS$_2$ systems

($\sigma_n \sim 10^{10}$ cm$^{-2}$)[14,30]. Moreover, the tunability of mobile charge density in 2D TMD system can be probably achieved in precisely manipulating the polarization field in supporting polycrystalline ferroelectric layer through setting different poling voltage, for example, the polarization in polycrystalline BFO can be changed from 5 to 70 μC cm$^{-2}$ thus would modulate the charge density as large as one order of magnitude[35,36].

**Direct verification of pn homojunction.** The spatially resolved spectral measurements provided clear evidence and insight into the ferroelectric control of WSe$_2$ carrier densities on a BFO substrate; a question arises whether one can reveal the rectification of a WSe$_2$ pn homojunction on reversed BFO domains. The fabrication of the WSe$_2$ pn diode (Diode-T) made use of the CVD growth of crystals of WSe$_2$, with a monolayered and triangle sheet (area $\sim 100$ μm$^2$) transferred onto a 30-nm-thick BFO substrate and then employed with the same ferroelectric switching in the AFM setup. Figure 4a presents a scanning electron microscope (SEM) image of the device used to measure the electrical transport measurements. In this way, the WSe$_2$ sheet became ferroelectrically doped into p-type (right half) and n-type (left half) conducting regimes to form a WSe$_2$ pn homojunction. All transport measurements were performed near 300 K and in

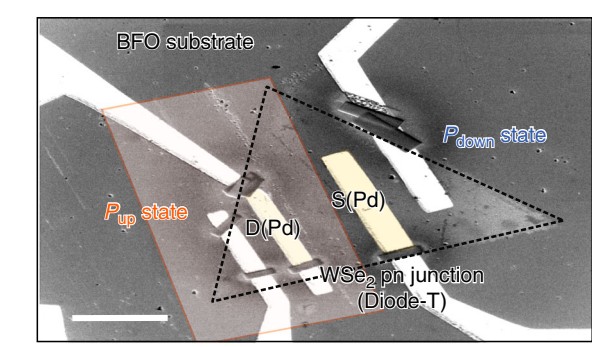

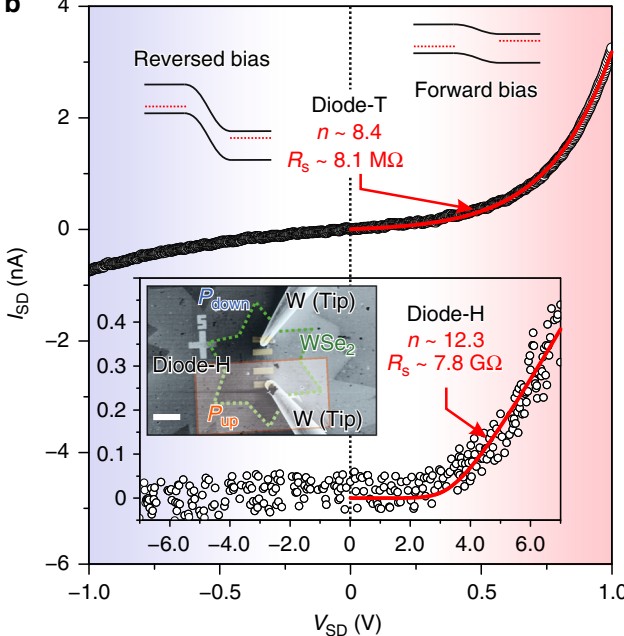

**Fig. 4** The SEM images of the devices for electrical transport measurements and their corresponding I–V curves. Measurements of electrical transport of a pn homojunction in WSe₂/BFO devices. **a** SEM image for a WSe₂ homojunction (Diode-T) from a top view of the junction. **b** Current measured as a function of voltage for the pn WSe₂ junctions on BFO (~30 nm, Diode-T) and thicker BFO (~60 nm, Diode-H) layers. The fits of the Shockley equation with extended series resistance at the forward-bias region give the ideality factor n and series resistance $R_s$ of WSe₂ diodes on thin (~30 nm, Diode-T) and thick (~60 nm, Diode-H) BFO layers. Both scale bars in the SEM images are 10 μm

vacuum, wherein the voltage bias $V_{SD}$ is applied and the direct current $I_{SD}$ is measured between source (S) and drain (D) contacts. Here, we choose to use Pd metal to form low-resistance ohmic contacts with WSe₂ because of its high work function[19]. It is worth noting that the Pd electrode is also found to conduct current into the BFO substrate when patterned upon it. Two conducting tungsten tips controlled by the nanomanipulator installed in SEM are utilized to conduct the electrical measurements. To prevent inevitable BFO substrate effect, the Pd metal electrodes are being cut off connections with BFO substrate using focused-ion-beam etching, which turning into isolated pad on the source and drain points (see Fig. 4a, b inset).

Figure 4b shows current measurements for the WSe₂ pn homojunction. Under reversed bias, the diffusion potential barrier height between the p-type and n-type sides becomes too high to flow a significant current through the junction and shows rectification behavior characteristic of a classic diode. Such diode

$I_{SD}$–$V_{SD}$ characteristics are typically modeled by the Shockley diode equation with extended series resistance $R_s$[37,38],

$$I_{SD} = \frac{nV_T}{R_s} W\left[\frac{I_0 R_s}{nV_T} \exp\left(\frac{V_{SD} + I_0 R_s}{nV_T}\right)\right] - I_0, \quad (1)$$

where $I_0$ is the reverse-bias saturation current and $R_s$ is the series resistance, while $nV_T$ is the thermal energy at room temperature with the ideality factor n ($n \geq 1$) of the diode. W is the Lambert W function. At the forward-bias voltages, the $I_{SD}$–$V_{SD}$ curve of our WSe₂ pn diode with a homo-interface is well modeled by the diode equation with $n \sim 8.4$, series resistance $R_s \sim 8.1$ MΩ. This ideality factor n is quite low compared with what have been observed for the CVD-grown TMD diodes, normally $n > 10$[39,40], however, it is still higher than the n value obtained in certain TMD diodes made by exfoliated natural crystal with high-quality[12,41,42]. Furthermore, no obvious reverse-saturation current is observed, indicating that current transmission through the whole WSe₂/BFO device is not limiting in the diode and suggesting that the BFO layer acts not as a good insulating layer to prevent current leakage as shown in Supplementary Figure 5.

To explore issues of current rectifying associated with the supporting BFO layer quality and to better understand the nature of current transmission through Pd metal contacts and WSe₂, we made another WSe₂ diode (Diode-H) consisting of a transferred CVD-grown WSe₂ monolayer flake having hexagram shape and large area (~700 μm²) on a thicker BFO layer (thickness ~ 60 nm) with opposite polarization states (shown in inset of Fig. 4b). Clearly, the current rectification can be reproducibly observed as shown in the $I_{SD}$–$V_{SD}$ characteristics. The low saturated current (~10⁻¹⁵ A, below the 1 pA noise level of the measurement) is extracted from diode equation fitting and can be observed at high reversed bias (0 ~ −5 V), confirming that the thick BFO layer indeed inhibits current leakage. The higher n (~12.3) value and larger series resistance $R_s$ (~7.8 GΩ) based on the fitting result with the turn-on point in the $I_{SD}$–$V_{SD}$ plots showing at large forward bias is attributed to a high series resistance consistently associated with the poor conductive quality of a hexagram shaped WSe₂ layer[43–46]. Moreover, on this large WSe₂ diode, it is easy to make both Pd electrodes on WSe₂ in $P_{down}$ states (pp junction) and in $P_{up}$ states (nn junction). Since Pd is a high work function metal[47], which means its Fermi-level will align valence band edge of WSe₂ for efficient hole injection, therefore, the pp junction shows a nearly ohmic $I_{SD}$–$V_{SD}$ relation at low $V_{SD}$. The $I_{SD}$–$V_{SD}$ relation in the nn junction shows non-linear due to the Schottky barrier formed at the n-type WSe₂/Pd interface, in which shown in Supplementary Figure 6. Apparently, minimizing the series resistance and defect densities including using asymmetric metal contacts on exfoliated crystalline WSe₂ should significantly improve the performance of this ferroelectricity-assisted WSe₂ pn diode.

## Discussion
To summarize, we designed and implemented a ferroelectrically controlled WSe₂ diode using a lateral pn homojunction without a biased electrode gate. Employing a monolayer WSe₂/BFO structure, we demonstrated non-volatile control of the band structure, surface potential, and light emission of monolayer WSe₂ with opposite polarization domains on a BFO substrate. With the quantitative analysis and modeling of current rectifying, as device quality improves, our gate-free 2D diode promises a comparable capability of current rectification with that of a conventional bias-gating-type 2D device. This evidence emphasizes the possibility of non-volatile control and provides an alternative functionality in ferroelectric doping of 2D materials, thus opening a wide vista of TMD-based quantum electronics and photonics.

## Methods

**Crystal growth**. A tungsten diselenide ($WSe_2$) sample was prepared with CVD. On evaporation of $WO_3$ at a high temperature, the $WO_3$ powders were filled in a home-made quartz reactor with a transfer tube of tunable length to enable a stable flow of $WO_{3-x}$ vapor that served as a reactant evaporated from a high-temperature zone to react with Se on a substrate surface. In the low-temperature zone, an individual $WSe_2$ monolayer was synthesized in a temperature range 650–750 °C. Monolayer $WSe_2$ typically formed a triangular shape in a monolayer thickness to transfer onto the BFO ferroelectric layer.

An epitaxial thin film (30 nm and 60 nm) of BFO was fabricated on the SRO-buffered STO (001) single-crystalline substrate via pulsed-laser deposition (KrF excimer laser, $\lambda = 248$ nm); the laser beam was focused on a $BiFeO_x$ ceramic target with energy density ~2.5 J cm$^{-2}$ and repetition rate 10 Hz. The samples were deposited at substrate temperature 700 °C under oxygen at pressure 100 mTorr. The samples were cooled to near 300 K under oxygen at pressure 1 atm after deposition of the thin film. The ferroelectric epitaxial BFO/SRO/STO(001) film possessed out-of-plane downward polarization; the domain structures as grown were confirmed with a PFM.

**Details of SPEM/S measurement**. The localized SR-PES technique provides a powerful method to obtain direct information about a band structure with varied polarization; soft X-rays (photon energy 400 eV) were used at the SPEM end station located at beamline 09 A1 of Taiwan Light Source in NSRRC. The soft X-ray beam focused by the Fresnel zone plate and the order-sorting aperture at the focal plane was about 100–200 nm in diameter. All measurements were undertaken near 300 K. The energy resolution was estimated to be better than 100 meV. Based on the SPEM images, the focused beam was movable to a specific location to record high-resolution PES of a microscopic area.

**Data availability**. The data that support the findings of this study are available from the corresponding authors upon reasonable request.

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

## Acknowledgements

We thank Mr. Yen-Chien Kuo and NSRRC staff for their skillful assistance during the synchrotron radiation experiments. Ministry of Science and Technology of Taiwan provided partial financial support of this work.

## Author contributions

C.-L.W. and C.-H.C. conceived the experiments. T.-M.C. coordinated the electrical transportation effort, S.-T.L, S.-C.H, T.-H.-Y.V, and C.-J.C. carried out the

measurements. Y.-C.C. coordinated the AFM-related efforts, Y.-D.L. and Y.-Y.C. performed the measurements. Y.-H.L. coordinated the TMD growth and X.-Q.Z. prepared the WSe$_2$ flakes. Y.-H.C. and J.-C.Y. prepared the BFO substrates. J.-W.C., S.-S.W. and Y.-X.C. carried out the SPEM measurements. J.-W.C., C.-H.C., and C.-L.W. analyzed the data, and wrote the manuscript with the inputs from all of the other co-authors.

## Additional information

**Competing interests:** The authors declare no competing interests.

