## [Peer Review File · Nature Communications]

Reviewers' comments:

Reviewer #1 (Remarks to the Author):

In this work, the authors propose electrically inducing p- and n-type regions in a single TMD material by using the polarization property of a ferroelectric. The concept is simple and interesting. However, the electrical characteristics of the p-n junction needs significant improvement.

Can the p- and n-doping concentrations be tuned by modulating the amount of polarization induced in the ferroelectric? One way to do this will be to traverse through the minor loops.

The authors state that the monolayer thickness of WSe₂ is about 1.8 nm (line 91). The monolayer thickness of WSe₂ should be around 0.7 nm.

It appears that the ferroelectric poling was done using a metal probe. Scanning systems are inherently slow. How efficient is this method for large substrates?

The leakage through the BFO layer is too large to accurately determine I_s .

The p-n junction diode suffers from very high series resistance and low current. The diode ideality factor without considering the series resistance (about 45 giga Ohms) is close to 10. The current through the diode is in the nA range (total resistance of about 0.5 to 1 giga Ohms). It is unclear how the series resistance is much higher than the total resistance.

Even in the off-state of the diode, the extracted series resistance is too high to accurately determine the ideality factor. What is the confidence (error) on the extracted ideality factor?

Reviewer #2 (Attached).

Reviewer #3 (Remarks to the Author):

General Comments

Novelty in claims?: Yes

Of Interest to the research community?: Yes

Will the paper influence thinking in the field?: Yes

Claims convincing?: More evidence needed

SCOPE AND CLAIMS:

1. One of the claims of the paper is that polarization in the BFO substrate can be used to realize fixed doping in TMDCs. The BFO substrate however is observed to have a significant conductance as seen in Fig. S5. In presence of this parallel substrate conductance, the practicability of this method is not clear, and should be addressed in the section describing the advantages of this method.

2. Line 91: The WSe₂ used is mentioned to be monolayer but its thickness is reported to be ~1.8 nm from AFM, which is at least 2 monolayers going by previous reports (e.g. Fang et al., Nano letters 12.7 (2012): 3788-3792); could the authors comment on the discrepancy? In line 90 it is mentioned that the "As in the case of the scanning line profile shown in Fig. 2a..."; however, the actual line scan showing the thickness of the WSe₂ is missing in Fig. 2a.

METHODS

1. The conductance of the BFO substrate looks significant from Fig. S5a. Could the authors comment on how this conductance is accounted for in the WSe₂ p-n junction I-V characteristics? In absence of isolation of this parallel conductance, it'd be erroneous to draw any strong

conclusion about the WSe2 I-V characteristics.

2. The rectification shown in Fig. 4B is less than a factor of 4 between forward and reverse bias currents for the bias range shown which raises the question if it's due to the p-n diode as claimed; typical p-n diode rectification ratios range in 4-5 orders of magnitude. Asymmetric $I_{(SD)}$ - $V_{(SD)}$ I-V characteristics are frequently seen in WSe2, MoS2, etc, due to the Schottky barriers at the contacts. Could the authors comment how the effects of the contacts are accounted for?

3. The characteristics for the p-p junction is shown in Fig. 4, but not the n-n junction; the I-V for the n-n junction should be included in Fig. 4. The argument for the presence of a p-n barrier would be more convincing if it can be shown that the current for the p-n case is lower than both the p-p and the n-n case.

4. In Line 206 a value of 7.3 pA is used for the diode reverse saturation current I_S ; however, the maximum reverse bias current of Fig. 4B is in close to 1 nA. Could the authors explain how the I_S is calculated?

5. There is likely a typo in the diode equation (line 204); accounting for the voltage drop across the series resistance R_S , the diode equation should be $I = I_S [\exp((V_{(SD)} - IR_S)/(\eta V_T)) - 1]$

6. In analyzing the p-n forward bias characteristics, the effect of the series resistance is not clear. For higher V_{DS} , where an ideality factor of 9.5 is found, the current $I_{(SD)}$ is still observed to follow $V_{(SD)}$ exponentially, i.e. in the log y scale, $I_{(SD)}$ follows $V_{(SD)}$ linearly.

if the current is indeed limited by the series resistance, then $I_{(SD)}$ becomes a linear function of $V_{(SD)}$ as described below (not an exponential one as in Fig. 4B):

When $V_{(SD)} - IR_S \gg V_T$, the equation above can be written as $I = I_S \exp((V_{(SD)} - IR_S)/(\eta V_T))$, or $\log_{10} [I] = \log_{10} [I_S] + (V_{(SD)} - IR_S)/(\eta V_T)$, or, $I = V_{(SD)}/R_S + (\eta V_T)/R_S \log_{10} [I_S/I]$, or $I \approx V_{(SD)}/R_S$ when $I_S \ll I$

7. In extracting the ideality factor of 1.1 in Fig. 4B, the considered I-V range is rather low, i.e. only a little more than one decade in current and 100 mV in the voltage, compared to what's typically reported for p-n junction analysis. Could the authors comment on why the ideality factor increases so rapidly after 100 mV ($\sim 3V_T$ at 300 K) when the p-n junction barrier height (built-in voltage) is ~ 450 meV (Fig 2c)? Usually the effect of the series resistance becomes important only when the S-D voltage becomes comparable to the p-n built-in voltage.

8. Line 214-216: It's unclear which series resistance is referred to as the dominant resistance. The resistance between the two metal electrodes (W and Pd) in contact, as done in all electrical probing systems is typically < 100 Ohms. It is highly unlikely that this resistance could account for any series resistance effect in Fig. 4a (total resistance in the GOhm range).

RESULTS

While the data from the physical characterization (binding energy spectrum) shows clear trends supporting the doping claims of the paper, the electrical data is weak. There are issues of isolating the parallel substrate leakage, effect of contact barriers, low rectification ratio, as listed above, which should be addressed in a revision.

Response to Referee 1

Reviewer's comments: In this work, the authors propose electrically inducing p- and n-type regions in a single TMD material by using the polarization property of a ferroelectric. The concept is simple and interesting. However, the electrical characteristics of the p–n junction needs significant improvement.

1. Can the p- and n-doping concentrations be tuned by modulating the amount of polarization induced in the ferroelectric? One way to do this will be to traverse through the minor loops.

Response:

We thank referee for the positive comments and insightful suggestions on our manuscript. In the ferroelectric film, the minor loop can only be obtained in a multidomain structure of the polycrystalline film. The ferroelectric BFO film in this study is epitaxial film having single crystal orientation, so the out-of-plane switchable components are all the same in the film. Therefore, in this study, we can choose only two states, P_{up} and P_{down} , to manipulate carrier concentration in TMD material. According to this criticism, the description about the charge density tuning TMD using polycrystalline ferroelectric substrate has been added on page 10 (1st paragraph and row 12).

2. The authors state that the monolayer thickness of WSe₂ is about 1.8 nm (line 91). The monolayer thickness of WSe₂ should be around 0.7 nm.

Response:

Thanks referee for your insightful questions. In a previous study, the monolayer thickness of WSe₂ was $d_{WSe_2} \sim 0.7$ nm which was made by mechanical exfoliation. And the monolayer thickness of WSe₂ obtained from $d_{WSe_2} \sim 1.1$ nm in WSe₂ which was made by CVD growth [1,2]. Here, we attribute this larger thickness to the presence of water molecules trapped at WSe₂/BiFeO₃ interface due to the slightly hydrophilic character of BiFeO₃ substrate [3,4]. In contrast to the monolayer thickness (~ 1.8 nm) measured by piezo-force microscopy (PFM) mode, we suggested the thickness measured by tapping-mode (~ 1.5 nm) is more accurate because of high sensitivity of cantilever oscillating using in WSe₂ on BFO system (See Fig. R1), therefore, we have changed the WSe₂ thickness from 1.8 nm to 1.5 nm in the revised manuscript on page 5 (2nd paragraph and row 8). Also, the monolayer WSe₂ clearly shows the transition in PL spectra of Fig. R2, which is in agreement with the magnitude of the direct bandgap ($E_g \sim 1.65$ eV) and thickness dependence of normalized PL spectra for different layers of WSe₂ [5].

- [1] A. S. Pawbake, M. S. Pauer, S. R. Jadkar and D. J. Late, *Nanoscale* **8**, 3008 (2016)
- [2] J. Quereda, A. Castellanos, N. Agrait and G. Rubio-Bollinger, *App. Phys. Lett.* **105**, 053111 (2014)
- [3] Y. C. Chen, C. -H. Ko, Y. C. Huang, J. C. Yang and Y. -H. Chu, *J. App. Phys.* **112**, 052017 (2012)

- [4] Y. Kim, C. Bae, K. Ryu, H. Ko, Y. K. Kim, S. Hong and H. Shin, *App. Phys. Lett.* **94**, 032907 (2009)
- [5] K. Xu, Z. Wang, X. Du, M. Safdar, C. Jiang and J. He, *Nanotechnology* **24**, 465705 (2013)

Figure R1: Height measurement for as-grown monolayer WSe₂ on BFO by AFM.

Figure R2: PL spectra of different layer WSe₂ [5] and as-grown monolayer WSe₂ using in the study.

3. It appears that the ferroelectric poling was done using a metal probe. Scanning systems are inherently slow. How efficient is this method for large substrates?

Response:

In this study, we made a *pn* homojunction in monolayer WSe₂ by using scanning probe technique to switch BFO polarization field with nanoscale precision. The efficiency of the scanning-probe assisted ferroelectric poling method is quite slow, in our case, for a typical

WSe₂ flake on BFO (dimension size around 10 μm), the consumption time is estimated into 12 mins for a scan of 10 × 10 μm² scanning area (Scan lines: 512 × 512 and scan rate: 0.7 line/s). To raise the efficiency, using a biased metal pads/electrode or a high-energy scanning electron beam could be achieved desired rate for polarization field poling on ferroelectric thin film [6].

[6] D. B. Li, D. R. Strachan, J. H. Ferris and D. A. Bonnell, *J. Mater. Res.* **21**, 935 (2006)

4. The leakage through the BFO layer is too large to accurately determine I_s .

Response:

Indeed, the leakage is quite large and no obvious reverse-saturation current is observed, since that current transmission over the WSe₂ *pn* homo-interface is not limiting in the diode and suggesting that the BFO layer acts not as a good insulating layer to prevent current leakage as shown in Fig. S5 (Supporting Information). Therefore, to explore issues of current rectifying associated with the supporting BFO layer quality, we made another WSe₂ diode on a thicker BFO layer (thickness ~ 60 nm). Clearly, the current rectification can be reproducibly observed and the low saturated current (~10⁻¹⁵ A, below the 1 pA noise level of the measurement) is extracted from diode equation fitting and can be observed at high reversed bias (0 ~ -5 V), confirming that the thick BFO layer indeed inhibits current leakage. According to this comment, we clarify the determination of I_s on page 13 (1st paragraph and row 7).

5. The p-n junction diode suffers from very high series resistance and low current. The diode ideality factor without considering the series resistance (about 45 giga Ohms) is close to 10. The current through the diode is in the nA range (total resistance of about 0.5 to 1 giga Ohms). It is unclear how the series resistance is much higher than the total resistance.

Response:

We realized that the series resistance is much higher than the total resistance, which casts a doubt on the current-voltage curve fitting. Therefore, we fitted the forward I_{SD} - V_{SD} characteristics for a WSe₂ *pn* diode with the Shockley diode equation, where cover the whole bias regions. We obtained series resistance (R_s ~8.1 MΩ) with a reasonable order, which shows a good agreement with the estimated series resistance in gated *pn* junction [B. W. H. Baugher *et al*, *Nature Nanotechnology* 9, 262-267 (2014)]. Also, we performed another WSe₂ *pn* diode on thicker BFO. This higher series resistance (R_s ~7.8 GΩ), which is based on the turn-on point of the fit in the I - V plots showing at large forward bias, is consistently associated with the poor conductive quality of a hexagram shaped WSe₂ layer. This reflecting the quality of hexagram-type WSe₂ from CVD growth is not comparable to triangle-type WSe₂. But this resistance value is in a reasonable range, which confirms validity of Shockley fitting function. According to this comment, we clarify the determination of I_s on page 13 (1st paragraph and row 7).

6. Even in the off-state of the diode, the extracted series resistance is too high to accurately determine the ideality factor. What is the confidence (error) on the extracted ideality factor?

Response:

As discussed in the above criticism, with the quantitative analysis and modeling of current rectifying on two different WSe₂ diodes, we found that the extracted ideality factor shows

highly dependence on the crystalline perfection of 2D material rather than the fitting error of I - V curve. Even through the I - V shows no leakage current and low reverse-bias saturation current, the ideality factor still shows relatively higher value in CVD-grown WSe_2 than mechanically exfoliated WSe_2 . This observation is in a good agreement with previous study for various TMD pn junctions [7-11]. And a comment on how the ideality factor depends on crystalline quality of 2D material is added **on page 12 (1st paragraph and row 12)**.

[7] B. W. H. Baugher, H. O. H. Churchill, Y. Yang and Pablo Jarillo-Herrero, *Nature Nanotechnology* 2014, **9**, 262.

[8] W. Yang, J. Shang, J. Wang, X. Shen, B. Cao, N. Peimyoo, C. Zou, Y. Chen, Y. Wang, C. Cong, W. Huang and T. Yu, *Nano Lett.* 2016, **16**, 1560.

[9] H. G. Shin, H. S. Yoon, J. S. Kim, M. Kim, J. Y. Lim, S. Yu, J. H. Park, Y. Yi, T. Kim, S. C. Jun and S. Im, *Nano Lett.* 2018, **18**, 1937.

[10] H. -M. Li, D. Lee, D. Qu, X. Liu, J. Ryu, A. Seabaugh and W. -J. Yoo, *Nature Communications* 2015, **6**, 6564.

[11] H. -J. Chuang, X. Tan, N. J. Ghimire, M. M. Perera, B. Chamlagain, Mark M. -C. Cheng, J. Yan, D. Mandrus, D. Tomanek and Z. Zhou, *Nano Lett.* 2014, **14**, 3594.

Response to Referee 2

Reviewer's comments: The paper reports on the formation of homojunction p-n diodes in WSe₂ using the different polarization states of ferroelectric BFO substrate. The pi phase shift in the polarization is used to create atomically sharp p- and n-doped regions. The authors claim ideal diode formation and high carrier doping densities. It is known that ferroelectric films can induce large carrier densities. The idea of using different polarization states to create a p-n diode is novel, and some of the results are convincing, however there are several issues that should be addressed:

1. Their KPM results show a potential difference of only 450mV (or 450meV of Fermi level shift) between the P_{up} and P_{down} regions. Is this sufficient to change the doping profile? Considering that the doping density is very large (see Q2) this result does not seem to be consistent with the formation of a p-n junction formation. For a material with a bandgap of ~1.6 eV, 0.45 eV change sufficient to switch from one heavily doped state to another. Furthermore, this result is not consistent with the results of Fig. 3 from PES where 1 eV of fermi level shift is shown.

Response:

We thank referee for his/her support on the publication of this manuscript. In this paper, the band structure of WSe₂ pn homojunction on P_{up} and P_{down} of ferroelectric BiFeO₃ film was mainly characterized by SPEM and showed a large shift of about 1 eV. This energy shift (same as the Fermi level shift) is larger than a surface potential difference between the P_{up} (n-type) and the P_{down} (p-type) regions of about 450 meV, which is measured by KPM. This is attributed to that the charge screening from surface adsorbate layer adsorbed on WSe₂/BFO device structure, which will lead to measure a reduced surface potential difference in KPM. The most commonly found adsorbate is water, which is studied in previous KPM measurements on ferroelectric BFO thin film [H. Sugimura *et al*, Appl. Phys. Lett. 80, 1459 (2002) and Y. -C. Chen *et al*, J. Appl. Phys. 112, 052017 (2012)].

2. They predict a carrier density of $\sigma_p = 2.28 \times 10^{13} \text{ cm}^{-2}$ and $\sigma_n = 2.28 \times 10^{15} \text{ cm}^{-2}$. I would think such high carrier density would result in degenerately doped semiconductor. In other words, the build in voltage of the p-n diode should be more than the bandgap! This is not consistent with the PES results – i.e. question #1 above. Can you explain? Perhaps the authors don't appreciate the magnitude of these numbers judging from their statement "This two-ordered carrier manipulation from $10^{13} \sim 10^{15}$ ". No, assuming the intrinsic carrier density is nearly zero because of the large bandgap (ni~0), you are changing the carrier density by 1E28, not just two orders. Yes, they differ by 2 orders of magnitude.

Response:

We thank referee for pointing out this misleading statement, "This two-ordered carrier manipulation from $10^{13} \sim 10^{15}$ " about comparing the carrier density manipulation in the

manuscript. As other criticisms about estimating the carrier density, we redo the estimation based on the two-dimensional density of state under a parabolic approximation for the band dispersion near the CB bottom and the VB top modeled in the effective mass of mobile carrier with Fermi-Dirac statistics. We addressed that the electron and hole carrier densities to $\sigma_n \sim 10^{13}$ and $\sigma_p \sim 10^{10} \text{ cm}^{-2}$, respectively, in the revised manuscript on page 9 (2nd paragraph and row 12). Furthermore, based on the intrinsic carrier density ($\sigma_i \sim 1.26 \times 10^3 \text{ cm}^{-2}$) of WSe₂, we found that the electron density tuning range ($\sim 10^{10} \text{ cm}^{-2}$) of WSe₂ is quite larger than the mobile hole accumulation range ($\sim 10^7 \text{ cm}^{-2}$) when the polarization in BFO layer is naturally P_{down}, which is in agreement with the results of the defect charge screening of naturally polarization field (P_{down}) in BFO proposed in previous results. And the huge tuning range of mobile electron in WSe₂ agrees satisfactorily with the order of surface bound charges of the BFO layer ($\sim 10^{14} \text{ cm}^{-2}$) and is higher than other TMD junction systems that was reported for elemental doping MoSe₂ ($\sigma_n \sim 10^{11} \text{ cm}^{-2}$) and lateral heterojunction WSe₂/MoS₂ systems ($\sigma_n \sim 10^{10} \text{ cm}^{-2}$).

3. A following question to #2: The P-V hysteresis is very symmetrical (S1). How come the corresponding carrier densities differ by two orders of magnitude? Also, can you estimate the transition region from P_{up} to P_{down}? Why would you not get band-to-band tunneling since the transition region seems sharp and the doping densities are very large? Why wouldn't you see an Esaki diode?

Response:

Thanks for referee's concern about the connection between P-V hysteresis loop and manipulated carrier density. In BFO film, the saturated P-V can be conveniently obtained by sufficient external bias such applied, then saturated spontaneous polarization of ferroelectric is estimated around of $P_{BFO} \sim 60 \text{ } \mu\text{C}/\text{cm}^2$ ($\sigma_{n/p} \sim 10^{14} \text{ cm}^{-2}$) as shown in Fig. S1. And the surface bound charges of the BFO layer can use to accumulate the mobile electron in WSe₂ with the $\sigma_n \sim 10^{13} \text{ cm}^{-2}$, as shown in the manuscript. About the transition region from P_{up} to P_{down} state, we can measure it around $\sim 1.8 \text{ } \mu\text{m}$ as shown in the KPM line profile (Fig. 2(c)). This distance is quite larger than the ferroelectric domain wall width ($\sim \text{nm}$), which is highly influenced by stray field from BFO ferroelectric film [Y. -C. Chen *et al*, J. Appl. Phys. 112, 052017 (2012)]. Therefore, we suggest this huge width will not result in Esaki diode formation in our WSe₂ *pn* homojunction. Moreover, in the Esaki diode, electron can travel across the p-n junction as a tunneling diode, which will show negative resistance characteristic in the IV curve. However, we do not see the negative resistance behavior in our IV curve. Thus, we think that our gate-free WSe₂ *pn* diode should not work like an Esaki diode.

4. In Fig 4b, *pp* and *pn* IV curves are shown. The authors should show the *nn* IV curves. The diode IV curve is very soft and does not show a marked rectification behavior.

Response:

Thanks for referee about his/her comments to our manuscript. Current-to-voltage measurement in *pp*, *nn* and *p-n* junctions play a significant role, which supports our band structure in SPEM. Therefore, we have added the current to voltage curve measurement about *pp* and *nn* junction of our new WSe₂ diode having hexagram shape and large area ($\sim 700 \text{ } \mu\text{m}^2$) in the supplementary Fig. S6. And a new *p-n* junction exhibits a clear rectification behavior

with low BFO leakage current and reverse-bias saturation current ($I_0 \sim 10^{-15}$ A) shown in the insert of Fig. 4 of the revised manuscript.

5. The diode equation is wrong. $I_s R$ should be IR .

Response:

Thanks for referee's instruction. We have modified from Schottky diode into Shockley diode with Lambert-W function. And a comment on the notation is added and displayed on the page 12 (1st paragraph and row 6).

6. I am unconvinced by the IV curve shown in Fig. 4. First, the authors should show the full $\log(\text{abs}(ID))$ vs V_{sd} for both the reverse and forward bias. This way, we can better assess the assertion that $n \sim 1$. It is hard for me to believe that the $n \sim 1$ region extends only about 0.1V. In most diodes, this region is less sensitive to the value of n . It seems to me the true diode behavior is the region with $n \sim 9.5$. This region is also exponential (not linear). I also don't understand where the 44 GOhm contact resistance came from? The pp resistance is about 1GOhm, which means that the nn resistance must be large. Please refer back to Q4.

Response:

We thank referee for his/her insightful comments. We realized that the ideality factor should be extracted from the fitting covering whole forward bias range, therefore, there was a better way for predicting diode performance with fitting $I_{SD}-V_{SD}$ curve. Here, we fitted the forward $I_{SD}-V_{SD}$ characteristics for a WSe_2 pn diode with the Shockley diode equation, where covers the whole forward bias regions. The full $\log(\text{abs}(I_{SD}))$ versus V_{SD} are fitted by Shockley diode with Lambert-W function and showed the ideal factor as $n \sim 8.4$ and series resistance $R_s \sim 8.1$ M Ω (See Fig. R1). The order of magnitude series resistance is reasonable and exhibiting a good agreement with previous study [M. A. Hughes *et al*, Appl. Phys. Lett. 103, 133508 (2013) and B. W. H. Baugher *et al*, Nature Nanotech. 9, 262 (2014)]. According to this comment, we revised the IV curve fitting of our WSe_2 diode on page 12 (1st paragraph and row 6).

Figure R1: In a $I_{SD}-V_{SD}$ characteristics fitting sketching on a logarithmic current scale in a WSe_2 /BFO device.

7. SI section 5 should be redone. I don't understand why bulk (3D) properties are used to estimate the 2D carrier densities. Multiplying the 3D carrier density by the thickness of WSe₂ to get the 2D carrier density is not the correct way to do this! The authors should do the exercise using 2D material properties using the well-known 2D density of states. Yes, many-body effects alter both of analysis but this is the standard way to calculate them.

Response: We thank referee for his/her instruction and insightful suggestion about carrier densities estimation of this manuscript. We modified our estimation and provided more information about estimating carrier density in the manuscript **on page 33 (1st paragraph)** to help readers easily understand this calculation, and the details were attached in the supplementary material.

Response to Referee 3

Reviewer's comments:

1. One of the claims of the paper is that polarization in the BFO substrate can be used to realize fixed doping in TMDCs. The BFO substrate however is observed to have a significant conductance as seen in Fig. S5. In presence of this parallel substrate conductance, the practicability of this method is not clear, and should be addressed in the section describing the advantages of this method.

Response:

We thank referee for the positive comments and insightful suggestions on our manuscript. Indeed, the leakage current of the BFO substrate casts a shadow on the practicability of the proposed gate-free *p-n* diode. Therefore, we fabricated another WSe₂ diode on a thicker BFO layer (thickness ~ 60 nm). Clearly, the current rectification can be reproducibly observed as shown in the *I-V* characteristics (inset of Fig. 4 (b)). The low saturated current ($\sim 10^{-15}$ A, below the 1 pA noise level of the measurement) is extracted from diode equation fitting and can be observed at high reversed bias (0 ~ -5 V), confirming that the thick BFO layer indeed inhibits current leakage. With this observation, we believe that the BFO substrate conductance is a factor that can be controlled.

2. Line 91: The WSe₂ used is mentioned to be monolayer but its thickness is reported to be ~1.8 nm from AFM, which is at least 2 monolayers going by previous reports (e.g. Fang et al., Nano letters 12.7 (2012): 3788-3792); could the authors comment on the discrepancy? In line 90 it is mentioned that the “As in the case of the scanning line profile shown in Fig. 2a...”; however, the actual line scan showing the thickness of the WSe₂ is missing in Fig. 2a.

Response:

Thanks referee for your insightful questions. In a previous study, the monolayer thickness of WSe₂ was $d_{\text{WSe}_2} \sim 0.7$ nm which was made by mechanical exfoliation. And the monolayer thickness of WSe₂ obtained from $d_{\text{WSe}_2} \sim 1.1$ nm in WSe₂ which was made by CVD growth [1,2]. Here, we attribute this larger thickness to the presence of water molecules trapped at WSe₂/BiFeO₃ interface due to the slightly hydrophilic character of BiFeO₃ substrate [3,4]. In contrast to the monolayer thickness (~1.8 nm) measured by piezo-force microscopy (PFM) mode, we suggested the thickness measured by tapping-mode (~1.5 nm) is more accurate because of high sensitivity of cantilever oscillating using in WSe₂ on BFO system (See Fig. R1), therefore, we have changed the WSe₂ thickness from 1.8 nm to 1.5 nm in the revised manuscript on page 5 (2nd paragraph and row 8). Also, the monolayer WSe₂ clearly shows the transition in PL spectra of Fig. R2, which is in agreement with the magnitude of the direct bandgap ($E_g \sim 1.65$ eV) and thickness dependence of normalized PL spectra for different layers of WSe₂ [5].

- [2] J. Quereda, A. Castellanos, N. Agrait and G. Rubio-Bollinger, *App. Phys. Lett.* **105**, 053111 (2014)
- [3] Y. C. Chen, C. -H. Ko, Y. C. Huang, J. C. Yang and Y. -H. Chu, *J. App. Phys.* **112**, 052017 (2012)
- [4] Y. Kim, C. Bae, K. Ryu, H. Ko, Y. K. Kim, S. Hong and H. Shin, *App. Phys. Lett.* **94**, 032907 (2009)
- [5] K. Xu, Z. Wang, X. Du, M. Safdar, C. Jiang and J. He, *Nanotechnology* **24**, 465705 (2013)

Figure R1: Height measurement for as-grown monolayer WSe₂ on BFO by AFM.

Figure R2: PL spectra of different layer WSe₂ [5] and as-grown monolayer WSe₂ using in the study.

3. The conductance of the BFO substrate looks significant from Fig. S5a. Could the authors comment on how this conductance is accounted for in the WSe₂ p-n junction I-V characteristics? In absence of isolation of this parallel conductance, it'd be erroneous to draw any strong conclusion about the WSe₂ I-V characteristics.

Response:

The I-V characteristics of our WSe₂ shown in Fig. 4 (a) has no obvious reverse-saturation current. This is because that the current transmission over the WSe₂ *pn* homo-interface is not limiting in the diode and suggesting that the BFO layer acts not as a good insulating layer to prevent current leakage as shown in Fig. S5 (Supporting Information). Therefore, we made another WSe₂ diode on a thicker BFO layer (thickness ~ 60 nm). Clearly, the current rectification can be reproducibly observed as shown in the *I-V* characteristics (inset of Fig. 4 (b)). The low saturated current (~10⁻¹⁵ A, below the 1 pA noise level of the measurement) is extracted from diode equation fitting and can be observed at high reversed bias (0 ~ -5 V), confirming that the thick BFO layer indeed inhibits current leakage. Therefore, indeed, a clear current rectifying in the WSe₂ I-V characteristics can be observed while a good isolation of parallel conductance is achieved in the WSe₂/BFO system.

4. The rectification shown in Fig. 4B is less than a factor of 4 between forward and reverse bias currents for the bias range shown which raises the question if it's due to the p-n diode as claimed; typical p-n diode rectification ratios range in 4-5 orders of magnitude. Asymmetric I_(SD)-V_(SD) I-V characteristics are frequently seen in WSe₂, MoS₂, etc, due to the Schottky barriers at the contacts. Could the authors comment how the effects of the contacts are accounted for?

Response:

Thanks for referee's concern about the contact of our *p-n* diode. To better understand the nature of current transmission through Pd metal contacts and WSe₂, we made another WSe₂ diode consisting of a transferred CVD-grown WSe₂ monolayer flake having hexagram shape and large area (~ 700 μm²) on a thicker BFO layer (thickness ~ 60 nm). Comparing these two different diodes, we found that the diode rectification performance is very sensitive to the quality of WSe₂ monolayer. Since that the CVD-grown WSe₂ monolayer flake has poor quality than that made by the exfoliated natural WSe₂ crystal, the rectification ratio of our WSe₂ diode would be smaller than the 2D devices that frequently made by exfoliating natural crystal. In addition, we use Pd as our metal contact materials on both *p*-type and *n*-type WSe₂. Pd is a high work function metal, which means its Fermi-level will easily align the valence band edge of WSe₂ for efficient hole injection, therefore, the *pp* junction shows a nearly ohmic *I-V* relation at low V_{SD}. As expected, the *I-V* relation in the *nn* junction shows non-linear due to the Schottky barrier formed at the *n*-type WSe₂/Pd interface, in which shown in Fig. S6 (Supporting Information). According to above results, apparently, minimizing the series resistance and defect densities including using asymmetric metal contacts on exfoliated crystalline WSe₂ should significantly improve the performance, such as ideality factor and rectification ratio, of this ferroelectricity-assisted WSe₂ *p-n* diode.

5. The characteristics for the p-p junction is shown in Fig. 4, but not the n-n junction; the I-V for the n-n junction should be included in Fig. 4. The argument for the presence of a p-n barrier would be more convincing if it can be shown that the current for the p-n case is lower than both the p-p and the n-n case.

Response:

The I-V curves of the *pp*, *nn*, and *pn* junctions on the new WSe₂ diode are included in the supplementary of Fig. S6 and shown in the following Fig. R3. As expected, we can observe that the current for the *p-n* junction is significantly lower than both the *pp* and *nn* junctions. This result confirms that the *p-n* homojunction in our WSe₂ diode dominates the current rectification.

Figure R3: Measurements of electrical transport of a *p-n*, *nn* and *pp* homojunctions in WSe₂/BFO devices.

6. In Line 206 a value of 7.3 pA is used for the diode reverse saturation current I_S ; however, the maximum reverse bias current of Fig. 4B is in close to 1 nA. Could the authors explain how the I_S is calculated?

Response:

We realized that the reverse-bias saturation current is much higher than the background current, which casts a doubt on the current-voltage transport result. To improve this, we performed the other thicker BFO with lower leakage current for our gate-free WSe₂ *p-n* diode. In the I_{SD} - V_{SD} characteristics of the new WSe₂ *p-n* diode (shown in Fig. 4b inset), a clear rectification without BFO leakage current and low reverse-bias saturation current $I_0 \sim 10^{-15}$ A below the 1 pA noise level of the measurement can be observed.

To help reader easily understand the current to voltage and fitting result easily, we have modified our manuscript from page 11 (2nd paragraph) to page 14 (1st paragraph).

7. There is likely a typo in the diode equation (line 204); accounting for the voltage drop across the series resistance R_S , the diode equation should be $I = I_S [\exp((V_{SD}) - IR_S)/(\eta V_T)) - 1]$

Response:

Thanks for your useful instruction and insightful suggestion. According to referee's comments, we have modified from Schottky diode equation into Shockley equation with series resistance in manuscript. And a comment on the new equation is added and displayed on page 12 (1st paragraph and row 6).

8. In analyzing the p-n forward bias characteristics, the effect of the series resistance is not clear. For higher V_{SD} , where an ideality factor of 9.5 is found, the current I_{SD} is still observed to follow V_{SD} exponentially, i.e. in the log y scale, I_{SD} follows V_{SD} linearly. If the current is indeed limited by the series resistance, then I_{SD} becomes a linear function of V_{SD} as described below (not an exponential one as in Fig. 4B): When $V_{SD}-IR_S \gg V_T$, the equation above can be written as $I=I_S \exp((V_{SD}-IR_S)/(\eta V_T))$, or $\log I = \log I_S + (V_{SD}-IR_S)/(\eta V_T)$, or, $I = V_{SD}/R_S + (\eta V_T)/R_S \log I_S/I$, or $I \approx V_{SD}/R_S$ when $I_S \ll I$

Response:

We thank referee for his/her insightful comments. We realized that the ideality factor should be extracted from the fitting covering whole forward bias range, therefore, there was a better way for predicting diode performance with fitting $I_{SD}-V_{SD}$ curve. Here, we fitted the forward $I_{SD}-V_{SD}$ characteristics for a WSe_2 pn diode with the Shockley diode equation, where covers the whole forward bias regions. The full $\log(\text{abs}(I_{SD}))$ versus V_{SD} are fitted by Shockley diode with Lambert-W function and showed the ideal factor as $n \sim 8.4$ and series resistance $R_S \sim 8.1 \text{ M}\Omega$ and reverse-bias saturation current $I_0 \sim 10^{-11} \text{ A}$ (See Fig. R4). Using the same fitting method, the ideal factor as $n \sim 12.3$ and series resistance $R_S \sim 7.8 \text{ G}\Omega$ are extracted for the new WSe_2 pn diode. Higher n value and high series resistance are associated with the crystalline imperfection of the CVD grown WSe_2 flake. Minimizing the series resistance and defect densities including using asymmetric metal contacts on exfoliated crystalline WSe_2 should significantly improve the performance of this ferroelectricity-assisted WSe_2 p-n diode.

Figure R4: In a $I_{SD}-V_{SD}$ characteristics fitting sketching on a logarithmic current scale in a WSe_2/BFO device.

9. In extracting the ideality factor of 1.1 in Fig. 4B, the considered I-V range is rather low, i.e. only a little more than one decade in current and 100 mV in the voltage, compared to what's typically reported for p-n junction analysis. Could the authors comment on why the ideality factor increases so rapidly after 100 mV ($\sim 3VT$ at 300 K) when the p-n junction barrier height

(built-in voltage) is ~ 450 meV (Fig 2c)? Usually the effect of the series resistance becomes important only when the S-D voltage becomes comparable to the p-n built-in voltage.

Response:

Thanks for your useful instruction and insightful suggestion. Now, we used Shockley diode with series resistance to fit the current-voltage curve as shown in the revised manuscript (on page 12 1st paragraph and row 6) with the ideal factor $n \sim 8.4$, series resistance $R_s \sim 8.1$ M Ω , and reverse-bias saturation current $I_0 \sim 10^{-11}$ A. Comparing two different WSe₂ diodes, for what may concern about error of ideality factor, we found that the ideality factor shows highly dependence on the crystalline perfection of 2D material rather than fitting error provided by fitting program. Even the I-V shows no leakage current and low reverse-bias saturation current in the new WSe₂ diode, the ideality factor still shows relatively higher value in the diode made by CVD growth than that made by mechanical exfoliated WSe₂ [6-10]. The comments on how the ideality factor depends on crystalline quality of 2D material is added on page 12 (1st paragraph and row 12).

[6] B. W. H. Baugher, H. O. H. Churchill, Y. Yang and Pablo Jarillo-Herrero, Nature Nanotechnology 2014, **9**, 262.

[7] W. Yang, J. Shang, J. Wang, X. Shen, B. Cao, N. Peimyoo, C. Zou, Y. Chen, Y. Wang, C. Cong, W. Huang and T. Yu, Nano Lett. 2016, **16**, 1560.

[8] H. G. Shin, H. S. Yoon, J. S. Kim, M. Kim, J. Y. Lim, S. Yu, J. H. Park, Y. Yi, T. Kim, S. C. Jun and S. Im, Nano Lett. 2018, **18**, 1937.

[9] H. -M. Li, D. Lee, D. Qu, X. Liu, J. Ryu, A. Seabaugh and W. -J. Yoo, Nature Communications 2015, **6**, 6564.

[10] H. -J. Chuang, X. Tan, N. J. Ghimire, M. M. Perera, B. Chamlagain, Mark M. -C. Cheng, J. Yan, D. Mandrus, D. Tomanek and Z. Zhou, Nano Lett. 2014, **14**, 3594.

10. Line 214-216: It's unclear which series resistance is referred to as the dominant resistance. The resistance between the two metal electrodes (W and Pd) in contact, as done in all electrical probing systems is typically < 100 Ohms. It is highly unlikely that this resistance could account for any series resistance effect in Fig. 4a (total resistance in the GOhm range).

Response:

Based on the new fitting method (Shockley diode equation with extended series resistance), we found that the series resistance can be extracted from the fitting about 8.1 M Ω , resulting from *p-n* diode (R_s) and contact resistance from electrode contacts (R_c), which is within a reasonable order and shows a good agreement to estimated series resistance in other *pn* junction [B. W. H. Baugher *et al*, Nature Nanotechnology 9, 262-267 (2014)]. In the new WSe₂ diode having larger area and hexagram shape, the series resistance extracted from the fitting is obtained about 7.8 G Ω , which is acceptable since the crystalline quality is quite poor in CVD grown WSe₂ having larger area and hexagram shape and apparently increased the resistance in WSe₂ *p-n* diode [11-14]. The comments on how the resistance depends on crystalline quality of 2D material is added on page 13 (1st paragraph and row 13).

[11] B. Liu, Y. Ma, A. Zhang, L. Chen, A. N. Abbas, Y. Liu, C. Shen, H. Wan and C. Zhou, ACS

Nano 2016, **10 (5)**, 5153.

[12] J. Chen, B. Liu, Y. Liu, W. Tang, C. T. Nai, L. Li, J. Zheng, L. Gao, Y. Zheng, H. S. Shin, H. Y. Jeong and K. P. Loh, Adv. Materials 2015, **27**, 6722.

[13] J. -K. Huang, J. Pu, C. -L. Hsu, M. -H. Chiu, Z. -Y. Juang, Y. -H. Chang, W. -H. Chang, Y. Iwasa, T. Takenobu and L. -J. Li, ACS Nano 2014, **8**, 923.

[14] C. M Smyth, R. Addou, S. McDonnell, C. L Hinkle and Robert M Wallence, 2D Materials 2017, **4**, 025084.

Reviewer #2 (Remarks to the Author):

1) The equation on pg 12 is still wrong!

It should read:

$$I_{SD} = I_0 \left(e^{\frac{(V_{SD} - I_{SD}R)}{nKT}} - 1 \right)$$

The analysis should reflect this also. Currently, the entire equation is wrong with a misplaced parenthesis.

2) The revised manuscript now shows the n-n characteristics as asked. The I-V characteristics clearly confirm the formation of back-to-back Schottky junctions. So, in addition to forming a p-n junction, there is a Schottky junction in series with the p-n junction.

So, the authors should explain which diode dominates? I still remain unconvinced that the p-n diode dominates. The authors are too selective in what they show in the manuscript compared to what they show in the rebuttal. For example, Fig. R3 (for reviewer #3) should be included in the manuscript. The fact is that the region where the p-n diode dominates over the n-n junction occurs at $V_{SD} > 1.5V$ in Fig. R3. But, the I-V curves shown in the manuscript are shown for $V_{SD} < 1V$, where according to Fig. R3, is dominated by the n-n junction. Compare for example Figures R3 and R4 for reviewer #3.

3) Based on observing $n \sim 8.4$, I would suggest you strike the following sentence on pg 12:
"This ideality factor n is remarkable compared with what have been observed for the CVD-grown TMD diodes."

There is nothing remarkable about observing such a large ideality factor.

2. Line 91: The WSe₂ used is mentioned to be monolayer but its thickness is reported to be ~1.8 nm from AFM, which is at least 2 monolayers going by previous reports (e.g. Fang et al., Nano letters 12.7 (2012): 3788-3792); could the authors comment on the discrepancy? In line 90 it is mentioned that the “As in the case of the scanning line profile shown in Fig. 2a...”; however, the actual line scan showing the thickness of the WSe₂ is missing in Fig. 2a.

Response:

Thanks referee for your insightful questions. In a previous study, the monolayer thickness of WSe₂ was $d_{\text{WSe}_2} \sim 0.7$ nm which was made by mechanical exfoliation. And the monolayer thickness of WSe₂ obtained from $d_{\text{WSe}_2} \sim 1.1$ nm in WSe₂ which was made by CVD growth [1,2]. Here, we attribute this larger thickness to the presence of water molecules trapped at WSe₂/BiFeO₃ interface due to the slightly hydrophilic character of BiFeO₃ substrate [3,4]. In contrast to the monolayer thickness (~1.8 nm) measured by piezo-force microscopy (PFM) mode, we suggested the thickness measured by tapping-mode (~1.5 nm) is more accurate because of high sensitivity of cantilever oscillating using in WSe₂ on BFO system (See Fig. R1), therefore, we have changed the WSe₂ thickness from 1.8 nm to 1.5 nm in the revised manuscript on page 5 (2nd paragraph and row 8). Also, the monolayer WSe₂ clearly shows the transition in PL spectra of Fig. R2, which is in agreement with the magnitude of the direct bandgap ($E_g \sim 1.65$ eV) and thickness dependence of normalized PL spectra for different layers of WSe₂ [5].

1. For the benefit of the reader, the authors should mention briefly in the manuscript the thickness measurement technique and the arguments to justify the layer number calculations.

5. The characteristics for the p-p junction is shown in Fig. 4, but not the n-n junction; the I-V for the n-n junction should be included in Fig. 4. The argument for the presence of a p-n barrier would be more convincing if it can be shown that the current for the p-n case is lower than both the p-p and the n-n case.

Response:

The I-V curves of the *pp*, *nn*, and *pn* junctions on the new WSe₂ diode are included in the supplementary of Fig. S6 and shown in the following Fig. R3. As expected, we can observe that the current for the *p-n* junction is significantly lower than both the *pp* and *nn* junctions. This result confirms that the *p-n* homojunction in our WSe₂ diode dominates the current rectification.

2. Figure S6 in the revised manuscript is currently missing the *pn* junction I-V; it should be the same as Figure R3 in the rebuttal letter.

Response to Reviewer #2

Reviewer comments:

1. The equation on pg 12 is still wrong! It should read:

$$I_{SD} = I_0 \left(e^{\frac{q(V_{SD} - I_{SD}R)}{nKT}} - 1 \right)$$

The analysis should reflect this also. Currently, the entire equation is wrong with a misplaced parenthesis.

Response:

The above equation is the expression of I - V characteristic of the pn junction with series resistance R . But this diode equation is implicit function that is not possible to apply into curve fitting with explicit form in terms of basic mathematical functions. By introducing the Lambert's W function, the Shockley diode equation shown in our manuscript had been frequently used to fit the I - V characteristics of a 2D pn device with series resistance R [see, for examples, Pablo Jarillo-Herrero *et al.*, *Nat. Nanotechnol.* **9**, 262 (2014), Li, *et al.*, *Nat. Nanotechnol.* **12**, 901 (2017), and Memaran *et al.*, *Nano Lett.* **15**, 7532 (2015)]. In the following derivation, we have showed these two equations are the same, one is represented with the Lambert W function, and the equation that reviewer #2 showed above is represented by the original format. Using this derivation, the presentation of our equation in the manuscript is ensured without any ambiguity.

Here is the derivation starting from reviewer's equation.

$$I_{SD} = I_0 \left(e^{\frac{(V_{SD} - I_{SD}R)}{nV_T}} - 1 \right); \text{ where } V_T = \frac{KT}{q}.$$

$$\Rightarrow I_{SD} + I_0 = I_0 e^{\frac{(V_{SD} - I_{SD}R)}{nV_T}}$$

$$\text{Multiply } \frac{R}{nV_T} \Rightarrow \frac{(I_{SD} + I_0)R}{nV_T} = \frac{I_0 R}{nV_T} e^{\frac{(V_{SD} - I_{SD}R)}{nV_T}}$$

$$\text{Multiply } e^{\frac{(I_{SD} + I_0)R}{nV_T}} \Rightarrow \frac{(I_{SD} + I_0)R}{nV_T} e^{\frac{(I_{SD} + I_0)R}{nV_T}} = \frac{I_0 R}{nV_T} e^{\frac{(V_{SD} + I_0 R)}{nV_T}}$$

If we let $Z = \frac{(I_{SD}+I_0)R}{nV_T}$, then the above equation can be written as

$$Ze^Z = \frac{I_0R}{nV_T} e^{\frac{(V_{SD}+I_0R)}{nV_T}} \equiv f(Z), \text{ if we set } y = f(Z),$$

then Z can be found by the inverse function of $f(Z)$, i.e. $Z = f^{-1}(y)$.

Then we introduce lambert W function is given by $W(y) = f^{-1}(y) = Z$, thus,

$$Z = \frac{(I_{SD}+I_0)R}{nV_T} = W(y) = W\left[\frac{I_0R}{nV_T} e^{\frac{(V_{SD}+I_0R)}{nV_T}}\right].$$

From the equation $\frac{(I_{SD}+I_0)R}{nV_T} = W\left[\frac{I_0R}{nV_T} e^{\frac{(V_{SD}+I_0R)}{nV_T}}\right]$, now, the I_{SD} can be described by

$$I_{SD} = \frac{nV_T}{R} W\left[\frac{I_0R}{nV_T} e^{\frac{(V_{SD}+I_0R)}{nV_T}}\right] - I_0.$$

This is exactly the form that we write in the manuscript to describe the fitting process of a *pn* diode including a series resistance R .

2. The revised manuscript now shows the n-n characteristics as asked. The I-V characteristics clearly confirm the formation of back-to-back Schottky junctions. So, in addition to forming a p-n junction, there is a Schottky junction in series with the p-n junction.

So, the authors should explain which diode dominates? I still remain unconvinced that the p-n diode dominates. The authors are too selective in what they show in the manuscript compared to what they show in the rebuttal. For example, Fig. R3 (for reviewer #3) should be included in the manuscript. The fact is that the region where the p-n diode dominates over the n-n junction occurs at $V_{SD} > 1.5V$ in Fig. R3. But, the I-V curves shown in the manuscript are shown for $V_{SD} < 1V$, where according to Fig. R3, is dominated by the n-n junction. Compare for example Figures R3 and R4 for reviewer #3.

Response:

The Figures R3 and R4 for reviewer #3 were obtained from completely different diodes with distinct shapes and qualities, thus, Reviewer #2 incorrectly compares these two figures (and the I-V curves shown in Fig. 4 of the manuscript) and remains unconvinced that the p-n diode dominates in the rectifying behavior. We think that the descriptions in our previous manuscript was confusing and apologize for any misunderstanding caused to Reviewer #2. We have now added a description (here we marked them as Diode-T and Diode-H) to clearly identify their difference and showed in the SEM images and I-V curves (Fig. 4) of these two different diodes.

Reviewer #2 is correct that the *IV* characteristic for the *nn* junction indicates the formation of back-to-back Schottky junctions. In the following *IV* plots (Fig. RR1), we compared the *pn* with *nn* junction on the same WSe_2 sample (Diode-H). It has been shown that the *pn* junction requires a higher forward bias to be turned on and has a much higher breakdown voltage in reverse bias in comparison with the *nn* junction. This result is a clear evidence that the *pn* junction dominates over Schottky one.

Last but not least, we hope to emphasize that the novelty and significance of our work are on the creation of gate-free monolayer WSe_2 *pn* junction through the control of ferroelectricity, which has been demonstrated using three distinct – optical, photoemission-spectroscopic, and electrical – methods. The existence of a small Schottky junction in series with the *pn* junction does not and should not take away from the significance and novelty of our work.

Figure RR1. Current vs. voltage measured in *pp*, *nn* and *p-n* junctions in the WSe_2/BFO system (Diode-H).

3. Based on observing $n \sim 8.4$, I would suggest you strike the following sentence on pg 12: “This ideality factor n is remarkable compared with what have been observed for the CVD- grown TMD diodes.”

There is nothing remarkable about observing such a large ideality factor.

Response:

Thanks reviewer #2 for pointing out this misleading sentence “This ideality factor n is remarkable compared with what have been observed for the CVD-grown TMD diodes”. According to reviewer #2’s suggestion, we have modified our description on page 12 (1st paragraph and row 14) by “This ideality factor n is **quite low** compared with what have been observed for the CVD-grown TMD diodes, normally $n > 10$ ”.

Response to Referee 3

Reviewer comments:

1. For the benefit of the reader, the authors should mention briefly in the manuscript the thickness measurement technique and the arguments to justify the layer number calculations.

Response:

We thank referee for the positive comments and insightful suggestions on our manuscript. According to reviewer’s suggestion, therefore, we have added the thickness measurement technique and the arguments to justify the monolayer WSe₂ in the revised manuscript on page 5 (2nd paragraph and row 7) and four more references (Ref. 25~28).

2. Figure S6 in the revised manuscript is currently missing the pn junction I-V; it should be the same as Figure R3 in the rebuttal letter.

Response:

Thanks referee for your insightful suggestions. According to reviewer’s suggestion, we have added the *pn* junction I-V in the Fig. S6 of the revised manuscript.